# High-Risk Human Papillomavirus Infection in Lung Cancer: Mechanisms and Perspectives

**DOI:** 10.3390/biology11121691

**Published:** 2022-11-23

**Authors:** Julio C. Osorio, Felipe Candia-Escobar, Alejandro H. Corvalán, Gloria M. Calaf, Francisco Aguayo

**Affiliations:** 1 Laboratorio de Oncovirología, Programa de Virología, Instituto de Ciencias Biomédicas (ICBM), Facultad de Medicina, Universidad de Chile, Santiago 8380000, Chile; 2 Advanced Center for Chronic Diseases (ACCDiS), Pontificia Universidad Católica de Chile, Santiago 8320000, Chile; 3 Instituto de Alta Investigación, Universidad de Tarapacá, Arica 1000000, Chile; 4 Universidad de Tarapacá, Arica 1000000, Chile

**Keywords:** papillomavirus, lung, cancer

## Abstract

**Simple Summary:**

A subset of human papillomaviruses (HPVs), so-called high-risk (HR)-HPVs are the causal agents of cervical, anogenital and a group of head and neck carcinomas. Additionally, HR-HPVs have been detected in extragenital tumors including in lung cancer, though their role in this heterogeneous group of malignancies remains controversial. In this review, we address the epidemiological and experimental studies regarding the role of HR-HPV in lung cancer, worldwide, and we propose potential mechanisms. The evidence suggests that HR-HPVs are involved in the development of a variable subset of lung carcinomas in both smoker and non-smoker subjects.

**Abstract:**

Lung cancer is a very prevalent and heterogeneous group of malignancies, and most of them are etiologically associated with tobacco smoking. However, viral infections have been detected in lung carcinomas, with high-risk human papillomaviruses (HR-HPVs) being among them. The role of HR-HPVs in lung cancer has been considered to be controversial. This issue is due to the highly variable presence of this virus in lung carcinomas worldwide, and the low viral load frequently that is detected. In this review, we address the epidemiological and mechanistic findings regarding the role of HR-HPVs in lung cancer. Some mechanisms of HR-HPV-mediated lung carcinogenesis have been proposed, including (i) HPV works as an independent carcinogen in non-smoker subjects; (ii) HPV cooperates with carcinogenic compounds present in tobacco smoke; (iii) HPV promotes initial alterations being after cleared by the immune system through a “hit and run” mechanism. Additional research is warranted to clarify the role of HPV in lung cancer.

## 1. Introduction

Lung cancer is the main cause of cancer deaths around the world, with there having been 1.8 million deaths in 2020 and an estimated nearly 2 million cases by 2022 [1]. Lung cancer is associated with multiple etiological factors including environmental, lifestyle and/or genetic ones [2]. Although this cancer is a heterogeneous disease, tobacco smoke is the most important etiological factor in most of the populations around the world [3]. Due to its heterogeneity, it can be sorted into two main groups with subsequent subclassifications: small-cell lung carcinomas (SCLCs) represent near of 15% of the lung cancer cases, with there being a tendency for this to be decreasing in all the populations [4]. SCLCs are characterized by a neuroendocrine origin, aggressivity and a poor outcome [5]. Non-small-cell lung carcinomas (NSCLCs) are the most common type of lung malignancy, contributing to ~85% of cases, and it is subclassified into three groups: squamous cell carcinoma (SQC), adenocarcinoma (AdC) and large cell carcinoma (LCC) [6]. 

A plethora of persistent viral infections have been detected in lung carcinomas worldwide, including high-risk human papillomaviruses (HR-HPVs) [7], Merkel cell polyomavirus (MCPyV) [8], Jaagsiekte Sheep Retrovirus (JSRV) [9], John Cunningham Virus (JCV) [10], and Epstein–Barr virus (EBV) [11]. However, the role of these viruses in lung cancer has not been clarified because the very variable viral presence of them in the populations, them having a low viral load and there being some bias in the detection methods. In addition, a high percentage of lung cancer is etiologically associated with tobacco smoking, which suggest that the viruses could have an oncogenic role in non-smokers. HR-HPVs have been detected in significant percentages of lung carcinomas worldwide, although the etiological role of this malignancy remains enigmatic. Due to HPV infections not being a sufficient condition for carcinogenesis, we addressed its potential cooperation with tobacco smoke. We focused on epidemiological and mechanistic evidence regarding a potential role of HR-HPVs in lung cancer. 

## 2. Human Papillomavirus: Classification, Structure and Replication Cycle 

Human papillomaviruses (HPVs) are members of the *Papillomaviridae* family which includes five genera: alpha-HPV (α-HPV), beta-HPV (β-HPV), gamma-HPV (γ-HPV), mu-HPV and nu-HPV [12]. The β-HPV, γ-HPV, mu-HPV and nu-HPV infect the cutaneous epithelia, while α-HPV infects both the mucous and cutaneous epithelia [13]. The α-HPV includes the high-risk (HR)-HPV, intermediate-risk (IR-HPV) and low-risk (LR)-HPV genotypes [14]. The HR-HPVs can be subdivided as carcinogenic (Group 1), probably carcinogenic (Group 2a) and possibly carcinogenic (Group 2b) by the International Agency for Research on Cancer (IARC). HPV consists of a non-enveloped particle of 55 nm in diameter with an icosahedral symmetry [15], whose genome is made of double-stranded DNA with about 8,000 nucleotides, and it holds eight open reading frames (ORFs), encoding for six early and two late proteins [16]. Furthermore, the non-coding long control region (LCR), located between the L1 and E6 genes regulates the viral gene expression program, and it is currently divided into three segments: the 5′ segment, the central segment and the 3′ segment [17,18]. Cellular transcription factors including Ying-yang 1(YY1), Nuclear-factor 1(NF1), Activator protein 1 (AP1), Glucocorticoid receptor (GR) and Progesterone receptor (PR), among others, can bind to the cognate site in this region for viral gene expression regulation. Additionally, viral E2 protein binding to cognate sites inside the LCR stimulate or to suppress the viral transcriptional activity with effects depending on different HPV types and variants [19,20,21]. The E region encodes all the non-structural E1, E2, E4, E5, E6 and E7 proteins, with them having major functions in viral replication and cell transformation [22]. On the other hand, the L region encodes for the major (L1) and minor (L2) structural proteins. The E2 gene encodes for a protein that can regulate the expression levels of the viral gene products [23]. Indeed, depending on the binding sites that are occupied by E2 in the LCR, it acts as a transcriptional repressor or activator [24]. Additionally, E2 recruits E1, a protein with helicase activity, at the viral replication origin for DNA replication [25,26], and finally, it transfers the viral genome to the daughter cells during host cell division [27]. The E4 gene encodes for the most abundant viral proteins, with it having functions in viral genome amplification and viral release [28]. The E5 gene encodes for an oncogenic protein that contributes to the productive state of the viral cycle and tumor progression [29], with an important role in the epidermal growth factor receptor (EGFR) activation and immune evasion [30]. Additionally, the E5 protein shows different host cellular targets that affect EGFR regulation mainly through the endosomal acidification process or by disrupting the EGFR ubiquitination in human foreskin keratinocytes [31]. In addition, the E5 protein has effects on the heavy chain of human histocompatibility antigen (HLA) and can interfere with the antigen presentation, which would derive from HR-HPV immune evasion [32]. E6 and E7 are known to be involved in cell cycle dysregulation, and they are responsible for viral genome replication [33]. The E6 and E7 oncogene expression by the episomal HR-HPV can contribute to tumor lesion progression [34]. The E6 oncogene mediates cell transformation, in part, by forming a complex with cellular E3 ligase E6-Association Protein (E6AP) to target p53 for degradation by the ubiquitin proteasome pathway [35]. Further, E6 increases the hTERT activity and inhibits interferon regulatory factor-3 transcriptional activity [36]. On the other hand, E7 is a small, dimeric protein that can bind and promote pRB degradation, which causes premature E2F releasing [37]. In addition, E7 mediates the activation of cyclin E and cyclin A, which are required for malignant transformation [38,39]. Additional interactions between the E6/E7 proteins and host partners are also involved in carcinogenesis [40]. 

HPV is an epitheliotropic virus, and the infectious cycle occurs in the stratified squamous epithelium [41]. In the cervical cells, HPV infects the basal epithelial cells of the squamocolumnar junctions through a microlesion in the host’s skin or mucosa, gaining direct access to the target cells [42,43,44]. Firstly, HPV binds to the heparan sulfate proteoglycans (HSPGs) which is located on the surface of the epithelial cells or basal lamina [45]. It has been established that after the HPV cell attachment, conformational changes occur in the capsid due to the proteolytic cleavage of the L2 protein [46], and cell entry is achieved by an endocytosis-mediated mechanism [47]. A virion rupture is accomplished due to the acidification of the endosome, revealing an L2-viral DNA complex [48] that is transported by actin microfilaments towards the Golgi apparatus [49] and nucleus [50]. The nuclear entry requires mitosis that allows the rupture of the nuclear membrane, thus facilitating the viral DNA association with chromatin [50]. After its entry into the nucleus, the viral genome stabilizes as an episome that is attached to the host genome through the E2 viral protein [51]. The transcription starts from the early promoter and occurs only from one viral DNA strand [52]. Most HPV transcripts are polycistronic [53], and the generated pre-ribonucleic acid (RNA) messengers (mRNA) are spliced into smaller mRNA pieces encoding different viral proteins [16,54]. During early infection, all of the early viral proteins (E1, E2 and E4–E7) are expressed [55]. The early polyadenylation signal (pAe) is blocked by E2, allowing there to be longer reads of the viral genome, ending at the late polyadenylation signal (pAl) and then expressing late mRNAs [56]. Most HPV infections are cleared by the immune system from 1 to 2 years [57], despite persistent HPV infections being a prerequisite for the development of a high-grade precancerous lesion [58]. Viral genome integration allows for E6 and E7 overexpression, which causes cell cycle deregulation and increases the genetic instability [59,60].

## 3. Epidemiology of HPV in Lung Cancer

Syrjänen (1972) described for the first time the presence of a condyloma-type lesion with the presence of koilocytosis in a bronchial SQC. This description turned out to be very similar to reports of cervical cancer lesions [61]. Additionally, Syrjänen (1980) described 104 bronchial SQCs, in which three different types of condylomatous lesions (papillomatous, flat and endophytic) were identified. Furthermore, the presence of a condyloma-type lesion with the presence of koilocytosis was reported [62]. Stremlau et al. (1985) reported one anaplastic carcinoma in the lung that contained HPV16 genotypes [63]. In 1989, Syrjänen et al. discovered the presence of HPV6/16 genotypes in bronchial SQCs by conducting in situ DNA hybridization (ISH) [64]. During the 1990s, different European countries reported HPV detections in lung carcinomas at different frequencies. In France, Béjui-Thivolet et al. (1990) reported HPV in seven out of forty-three (16%) tumor specimens by using the same method [65], while Thomas et al. (1995) found HPV in five out of thirty-one (16%) cases by using a nested polymerase chain reaction (PCR) [66]. In Greece, Noutsou et al. (1996), reported an HPV presence in 15 out of 99 (15%) lung carcinomas by using PCR [67]. Conversely, in Germany, Welt et al. (1997) did not detect HPV in 38 specimens by using ISH/PCR [68]. In Greece, Papadopoulou et al. (1998) found HPV in 36 out of 52 (69%) lung carcinomas by using PCR/Southern blotting (SB). Interestingly, HPV16/18 were the most prevalent genotypes (37.9%) [69]. However, in the same country, Gorgoulis et al. (1998) reported the absence of HPV in 91 lung SQCs by using PCR/ISH. [70]. In Norway, Henning et al. (1998) found HPV in 37 out of 75 (49%) lung carcinomas from women by using PCR/ISH [71]. In Asia, different HPV frequencies in lung carcinomas were reported in the same decade. In Beijing in the Republic of China, Xing et al. (1993) found HPV in seven out of forty-nine (14.3%) tumor samples by using multiplex PCR [72], though Szabó et al. (1994) reported the absence of HPVs (6, 11, 16, 18, 31, 33, 52b and 58) in 47 lung cancers from Japan by using PCR [73], and Tsuhako et al. (1998) detected HPV in 19 out of 207 (9%) lung carcinomas by using PCR and non-isotopic ISH in the same country [74]. Liu et al. (1994) found five out of forty-nine (10%) HPV-positive lung carcinomas from China by using PCR/ISH [75]. In the same country, Hu et al. (1997), found HPV in 16 out of 50 (32%) lung carcinomas by using PCR/dot-blot hybridization [76]. In the US, two studies were reported in the 1990s. Yousem et al. (1992), detected HPV in seven out of fifty-eight (12.1%) lung cancers [77], and Bolhmeyer et al. (1998) reported an HPV presence in six out of thirty-four (18%) lung carcinomas by using PCR. HPV18 was detected in 6% of the samples (2/34) [78]. During the 2000s, variable frequencies of HPV in lung carcinomas were found. In France, Clavel et al. (2000) found HPV in five out of one hundred and eighty-five (2.7%) samples by the a Hybrid Capture II assay (HCA) [79], while in Poland, Miasko et al. (2001) reported an HPV presence in four out of forty (10%) NSCLCs [80]. In Turkey, Kaya et al. (2001) detected HPV in three out of twenty-six (11.5%) cases (primary SQC) by using NISH [81], and Zafer et al. (2004) found HPV in two out of forty (5%) lung carcinomas by using ISH, PCR and SB [82]. In France, Brouchet et al. (2005) reported no HPV presence in 122 specimens by using IHC/ISH [83], and Coissard et al. (2005) found HPV in four out of two hundred and eighteen (2%) lung tumors by using reverse line blotting (RLB), though the E6 mRNA was undetectable [84]. In Italy, Ciotti et al. (2006) found HPV in eight out of thirty-eight (21%) NSCLCs by observing the E6/E7 expression [85], and Giuliani et al. (2007) found HPV in 10 out of 78 (13%) lung carcinomas by using PCR and RFLP. Interestingly, the E6 and E7 transcripts were expressed in nine samples [86].

In Asia, the HPV frequency in the lung cancer samples were relatively higher than they were in other regions of the world. For instance, in Taiwan, Cheng et al. (2001) found HPV in 77 out of 141 (54.6%) lung carcinomas and in 16 out of 60 (26.7%) non-cancer control samples [87]. Additionally, Hsu et al. (2009) found the HPV16 E6 oncoprotein in 49 out of 217 (22.6%) and HPV18 E6 in 31 out 217 (14.3%) patients with stage I NSCLC by using immunohistochemistry (IHC) [88]. In China, Yu et al. (2009) found HPV in 43 out of 109 (39%) lung carcinomas by using INNO-LIPA PCR and reverse transcriptase (RT)-PCR [89], while Xu et al. (2009) reported an HPV16/18 presence in 32 out 44 of (73%) SQCs by using ISH. Integrated forms of HPV were found in twenty-three (52.27%) and episomal forms were found in nine (20.45%) of the cases [90]. In the same country, Zhang et al. (2009) reported an HPV16 (E6 and E7 oncogenes and L1 gene) presence in 18 out of 104 (17.3%) lung carcinomas by using PCR/sequencing [91]. In addition, Fei et al. (2006) found HPV in 32% of the 73 lung tumors by using ISH [92], and Wang et al. (2008) reported an HPV prevalence in 23 out of 313 (39.2%) fresh lung tumor specimens by using PCR and NISH [93]. In Okinawa in Japan (2000), Miyagi et al. reported an HPV presence in 78 out of 434 (18%) surgically resected lung tumors by using PCR/SB [94]. In Seoul in Korea, Park et al. (2007) reported an HPV presence in 58 out of 112 (52%) NSCLCs. HPV16, 18, and 33 were detected in 12 (10.7%), 11 (9.8%), and 37 (33.0%) of the cases, respectively [95]. In Tehran in Iran, Nadji et al. (2007) found HPV in 37 out of 141 (26%) lung carcinomas by using nested PCR [96]. In Singapore, Lim et al. (2009) reported no HPV presence in 110 patients with adenocarcinoma by using ISH [97]. In Latin America, some studies have detected HPV in lung carcinomas. Castillo et al. (2006) reported an HPV presence in 10 out of 36 (28%) of tumors which were collected from Colombia, Mexico and Peru by using PCR/SB [98], while Aguayo et al. (2007) detected the HPV presence in 20 out of 69 (29%) lung carcinomas from Chile by using PCR/SB. HPV16 was found in 11 cases (16%) [99].

In Europe after 2010, a high level of variation in the HPV frequency in lung cancer patients has been reported. In Croatia, Branica et al. (2010) reported the HPV (16, 18 and 33 genotype) presence in three out of eighty-four (3.6%) bronchial aspirates of lung carcinoma patients by using PCR [100]. In Italy, Carpagnano et al. (2011) reported an HPV presence in 12 out of 89 (13.5%) exhaled breath condensates (EBC) of lung cancer patients by using PCR/pyrosequencing [101]. In Finland, Syrjänen et al. (2012) found HPV in four out of seventy-seven (5,2%) lung carcinomas by using HPV genotyping with the Luminex-based Multimetrix. Three specimens were HPV16-positive, and one case was coinfected with HPV6/HPV16 [102]. In Italy, Galvan et al. (2012) reported absence of HPV in 100 lung carcinomas by using PCR and hybridization using specific probes [103]. In Greece, Sarchianaki et al. (2014) found HPV in 19 out of 200 (9.5%) lung carcinomas by using qPCR. The HPV genotypes more frequently detected were HPV16 (42.1%) and HPV11 (15.8%) [104]. In Greece, Argyri et al. (2017) found HPV E6/E7 mRNA in two out of sixty-seven (3%) lung tissues by using PCR and Real-time nucleic acid sequence-based amplification (NASBA). E6/E7 mRNA was not found in any of the tissue samples [105]. In Copenhagen in Denmark, Shikova et al. (2017) reported an HPV16/18 presence in 33 out of 132 (25%) lung carcinomas by using PCR with GP5+/6+ primers and a type-specific (TS) primers for HPV16/18. When the HPV was analyzed by conducting a consensus PCR, only five out of one hundred and thirty-two (3.8%) samples were HPV-positive [106]. In the Czech Republic, Jaworek et al. (2020) did not detect HPV in 80 primary NSCLCs by using qPCR [107]. In Bordeaux, France, Chaussade et al. (2022) reported HPV incidence rate ratios (IRR) of 1.8 [1.4–2.2] in a cohort of HIV-positive patients with lung cancer (incidence rates for 100 000 people per year) [108]. In Badalona, Spain, Sirena et al. (2022) found HPV in 23 out of 41 (56%) lung carcinomas by using qPCR. Furthermore, the HR-HPV types were detected in 16 out of 41 samples (39%, 95% CI 26–54%), and HPV16 was the most prevalent genotype [13/16 (81.3%, 95% CI 57.0-93%] [109].

After 2010 in Asia, the HPV frequency in the lung cancer samples were relatively higher than they were in other regions of the world. In Kagoshima in Japan, Baba et al. (2010) found HPV in nine out of thirty (30%) AdCs and in two out of twenty-seven (7%) SQCs by using a PCR/INNO-LiPA assay. This difference was statistically significant (*p* = 0.044). Interestingly, in 20 additional specimens from patients who were treated with gefitinib, HR-HPV was found in six out of eight (75%) lung AdCs from patients with a complete or partial response to gefitinib, and it was not detected in the twelve patients that did not respond to gefitinib [110]. In the same country, Iwakawa et al. (2010) reported absence of HPV 16/18/33 in 297 lung AdCs by using multiplex PCR [111]. Additionally, Aguayo et al. (2010) found HPV-16 in eight out of sixty (13%) of lung carcinomas from China, Pakistan and Papua New Guinea by using RT-qPCR. Additionally, HPV16 was detected in eight out of eighteen (44%) SQCs [112]. In Japan, Goto et al. (2011) found HPV in 20 out of 304 (6.6%) of lung cancer samples by using PCR/ISH. The HPV16/18 genotypes were present in the most (75.7%) of the patients with lung cancer [113]. In Iran, Jafari et al. (2013) found HPV in nine out of fifty (18%) lung SQCs by using nested PCR/sequencing [114]. In China, Yu et al. (2013) found HPV16/18 in 44% of the lung cancers by using PCR, the INNO-LIPA genotyping system and SB. An HPV infection was more prevalent in the SQCs (59.8%) when they were compared to the AdCs (17.5%). HPV16/18 were the most frequently detected type [115]. In China, Fan et al. (2015) reported an HPV presence in 22 out of 262 (8.4%) lung carcinomas by using PCR, ISH (p53 and p16 expressions) and a reverse dot blot. The HPV infection rates in the SQC cases were significantly higher than they were in the AdC cases (12.69 versus 3.91%) [116]. In China, Yu et al. (2015) found HPV in 100 out of 180 (55.6%) lung carcinomas by using PCR and IHC. HPV16 was detected in 67 out of 180 (37.22%) cases, and HPV18 was detected in 56 out of 180 (31.11%) cases [117]. In Iraq, Al-Shabbani (2015) reported an HPV presence in 30 out of 50 (60%) lung carcinomas by using ISH. HPV16 and HPV18 were detected in 32% and 28% of the samples, respectively [118]. In Hefei in the Republic of China, Li et al. (2016) found HPV in 27 out of 95 (28.4%) advanced-lung AdCs by performing the PCR amplification of a fragment of the HPV L1 gene [119]. In Fuzhou in the Republic of China, Xiong et al. (2016) reported HPV presence in seven out of eighty-three (8.4%) lung carcinomas by using PCR and reverse hybridization [120]. In Fuzhou in the Republic of China, He F et al. (2020) found HPV in 13 out of 140 (9.3%) lung cancer tissues by using PCR amplification (L1 primers) and specific probe reverse hybridization [121]. In Tehran-Iran, Rezaei et al. (2020) found HPV in 54 out of 102 (52.9%) lung cancers by using PCR for the L1 and E7 genes and by genotyping by INNO-LiPA [122]. In Iran, Hussen et al. reported an HPV presence in 61 out of 109 (56%) lung cancer tissues by using PCR and RLB [123]. In China, Wu et al. (2021) detected the presence of HPV DNA in 16 out of 100 (16%) patients with NSCLC by using RT-qPCR [124]. In Shenyang, Republic of China, Zou et al. (2021) found HPV in 183 out of 310 (59%) lung carcinomas by using qPCR. Furthermore, the expression levels of E6 mRNA and E7 mRNA in the SCLC group were significantly higher than those in benign cell group [125]. In Taiwan, Huang et al. (2022) reported an HPV incidence rate of 39.44 (37.47–41.52) (crude incidence rate, per 100,000 people per year) in lung cancer patients. Furthermore, it was concluded that an HPV infection was associated with the occurrence of adenocarcinoma of the lungs in both men and women (1.714; 1.572–1.870) [126].

After 2010, new studies were reported in America. In Mexico, Badillo-Almaraz et al. (2013) reported an HPV presence in 16 out of 39 (41%) lung cancer tissues by using PCR [127]. In USA, Joh et al., (2010) reported an HPV presence in five out of thirty (16.7%) lung carcinomas by using PCR and DNA sequencing. Only HPV16 and HPV11 were identified [8]. In the same country, Koshiol et al. (2011) reported no HPV presence in 399 lung carcinomas by using RT-qPCR [128]. In Canada, Yanagawa et al., 2013, reported HPV incidences in five out of three hundred and thirty-six (1.5%) lung cancers by using ISH and PCR [129], while Chang et al. (2015) reported negative results in 196 lung cancer samples by conducting ISH, an HR-HPV E6/E7 RNA detection and p16 IHC [130]. In the USA, Colombara et al. (2015) evaluated the HPV antibodies in 200 lung cancer cases and matched them to the controls. There was no evidence for a positive association between an HPV16 or 18 infection and the incidence of lung cancer [131]. In Brazil, de Oliveira et al. (2018) found HPV in 33 out of 63 (52%) lung carcinomas by using PCR, genotyping and IHC for E6 and E7. HPV16 was present in 27 out of 33 (81%) cases, and HPV18 was present in six out of thirty-three (19%) cases [132], while Silva et al. (2019) did not find HPV in 77 NSCLC patients by using PCR [133]. Table 1 shows the frequency of HPV detection worldwide since 1972.

## 4. HR-HPV-Mediated Lung Carcinogenesis: Potential Mechanisms

It has been suggested that both the natural and the iatrogenic squamous–columnar junctions (SCJs), which are present in the upper respiratory tree, are the predilected entry site of HPV in this anatomical region [7]. Interestingly, this is in line with a recent meta-analysis which showed that HPV is most frequently detected in lung SQCs than in AdCs worldwide [134]. Despite the molecular mechanisms of HR-HPV-mediated cervical and head and neck carcinogenesis having been widely studied [135], those which may be related to lung cancer are not clear. However, the studies suggest that the HR-HPV genome integration with the subsequent E6 and E7 overexpression are important events [99,136], with a subsequent p53 and pRB downregulation occurring, which in turn results in loss of apoptosis and cell proliferation, respectively [137]. HPV integration into the host genome is a random event that is not part of the normal HPV life cycle and whose mechanisms have not been completely understood. Indeed, the integration sites into the host genome have been mapped in cervical and head and neck cancer [138,139], although according to our knowledge, the studies addressing HPV integration sites in lung cancer are lacking. HPV genome integration can alter the expression levels of the genes near the integration sites. These altered genes in the “hotspots” show diverse roles including angiogenesis, cellular differentiation, migration, invasion, proliferation, and regulation of cell death, among others [140]. Of note, HPV integration is facilitated by repair processes that are activated in the cells with chromosomal instability, which is a hallmark of human cancer [141,142]. The “looping” model of HPV integration is the most accepted one. In this model, HPV integration is mediated by DNA replication and recombination, resulting in DNA concatemers, such as those which occur in SiHa and CasKi cervical cancer cells [143,144]. 

Tobacco smoke is the most important carcinogen involved in lung carcinogenesis worldwide [3]. In this context, the role of HPV needs to be clarified (Figure 1). Some groups in Asia have suggested that HPV works as a carcinogen in non-smoker subjects (Figure 1) [145]. Regarding these data, the signaling pathways have been characterized, and the potential biomarkers have been suggested. For instance, a frequent HR-HPV presence has been found in non-smoker, Taiwanese female lung cancers. Additionally, a significant increase in CDKN2A (p16) promoter hypermethylation was found in this group when it was compared to the non-smoker male lung cancers, suggesting that HR-HPV is potentially involved in this epigenetic alteration. Consecutively, it was demonstrated that this effect may be linked with the expression of the DNA methyltransferase 3b (DNMT3b) protein. Interestingly, it has been described an important role for the E6 oncoprotein in lung cancer. Indeed, E6 expression in lung tumors have shown to be associated with the tissue inhibitor of metalloproteinase 3 (TIMP-3) loss by promoter hypermethylation, thus inducing interleukin 6 production (IL-6). In addition, the expression of PD-L1 by the E6 oncoprotein through the ERK-C/EBPβ-TLR4-NF-κB signaling pathway promotes tumor growth and the invasiveness of lung carcinomas [146]. Of note, E6 can promote a reduction of microRNA-184, conferring cisplatin resistance in lung cancer via increasing Bcl-2 [147]. The same effect is promoted through cIAP2 upregulation via the EGFR/PI3K/AKT pathway by E6 in HPV16/18-positive lung cancer subjects [148]. Moreover, the induction of FOXM1 by the E6 oncoprotein through the MZF1/NKX2-1 axis may be involved in HR-HPV-mediated lung cancer progression and poor outcomes in HPV-positive patients [149]. An additional effect of E6, but not of E7, is to inhibit the antitumor activity of LKB1, a serine-threonine protein kinase, in lung cancer cells by downregulating the expression of kinesin family member 7 (KIF7) [150].

Cheng YW et al. showed that the HPV E6-induced promoter hypermethylation of the XRCC3 and XRCC5 DNA repair genes promoted increased benzopyrene (B[a]P)-induced DNA adducts, in turn, contributing to lung tumorigenesis. B[a]P is a polycyclic aromatic hydrocarbon (PAH) that is a recognized as a class I carcinogen that is present in tobacco, industrial exhaust, and environmental contamination. Additionally, arsenic [151], radon [152] and others environmental factors are etiologically related to lung cancer. For instance, it is known that tobacco smoking and second-hand smoke are etiologically related to 70–90% of lung cancer cases worldwide, though this percentage varies depending on the histological type and the sociodemographic factors that are involved [153,154]. In fact, tobacco smoke is a class I carcinogen whose oncogenic mechanisms and involved signaling pathways have been extensively understood in the past years [155]. Whether environmental compounds such as tobacco smoke can cooperate with HR-HPV for lung tumorigenesis is a controversial topic. Interestingly, smoker women who are infected with HR-HPV are more susceptible to cervical cancer when they are compared to non-smokers [156]. However, such a relationship has not been established in lung or head and neck cancers. Our group addressed a potential cooperation between HR-HPV and the cigarette smoke components in the models of lung epithelial cells. Additionally, we previously proposed the possibility that HR-HPV can cooperate with tobacco smoke for lung carcinogenesis using in vitro approaches. Muñoz et al. suggested that tobacco smoke increases the tumorigenic properties of E6/E7-expressing lung cells [157]. Additionally, our team demonstrated that tobacco smoke can activate the HPV16 early (p97) promoter in the lung epithelial cells, thus increasing the levels of the E6 and E7 oncoproteins. These oncoproteins can cooperate with tobacco smoke to increase the DNA damage in the lung epithelial cells [158]. In addition, other studies have established the molecular mechanisms of cooperation between HPV and tobacco in different cell contexts. In fact, Wei et al. reported that tobacco smoke extracts increase the E6/E7 levels only when the virus is found in an episomal state in the cervical cells, suggesting that tobacco smoke affects the HPV gene expression only in preneoplastic lesions before cancer development [159]. Local immunosuppressive effects, leading to the persistence of an HPV infection has been suggested [160,161,162]. Molecular alterations in HPV-associated lung cancers are summarized in Figure 2. 

Importantly, the viral load in lung tumors can shed light in respect to its role in this malignancy. Indeed, the very low viral load in lung carcinomas (less than 1 copy/cell) suggest the possibility that HPV is a mere bystander in the clinical samples. However, we cannot deny the possibility of a “hit and run” mechanism, by which the viral infection is involved in the initial events in carcinogenesis, and then the virus is cleared as it is no longer required for the cancer progression. Indeed, previous evidence shows that HPV can work by this “hit and run” mechanism [163,164,165]. Future research that is focused on lung preneoplastic lesions can elucidate this possibility.

## 5. Conclusions

An etiological role of HR-HPV in a subset of lung cancers including both AdCs and SQCs is plausible. However, there is a high variability in HPV detection rates among different countries, and moreover, in the same countries (from 0 to 100%). Probably, the factors related to the differential sensitivity among the detection methods can account for these differences because we lack a gold standard method for HPV detection in human tissues. Additionally, ethnic, lifestyle and sociodemographic factors can be involved in explaining such differences. The molecular mechanisms by which HPV promotes lung cancer are not fully known. As in cervical cancer, HR-HPV E6/E7 overexpression by HPV genome integration is an important event, though the completion of additional studies is necessary to address the consequences of such events for lung epithelial cell transformation. Thus, the analysis of some viral parameters such as the HPV copy number and the viral physical status in the lung clinical specimens is warranted in future studies. Because a “hit and run” mechanism cannot be denied, studies including the preneoplastic lesions of lung cancer will be valuable for the further characterization of this virus in this malignancy. As an HR-HPV infection is not a sufficient condition for carcinogenesis, the role of additional co-factors such as xenobiotics, including environmental factors (tobacco, arsenic, pollution, etc.) which can potentially cooperate with HPV for lung tumorigenesis remains an interesting point for future research.

## Figures and Tables

**Figure 1 biology-11-01691-f001:**
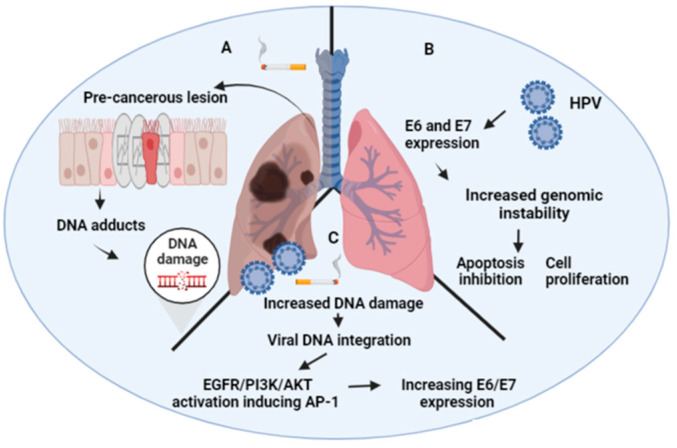
Tobacco smoke and HPV in lung cancer. (**A**) Tobacco smoking is the most important independent etiological factor of lung cancer. (**B**) HPV can be related to lung cancer as an independent carcinogen in non-smoker subjects. (**C**) HPV can cooperate with tobacco smoke for lung tumorigenesis. Designed using BioRender.

**Figure 2 biology-11-01691-f002:**
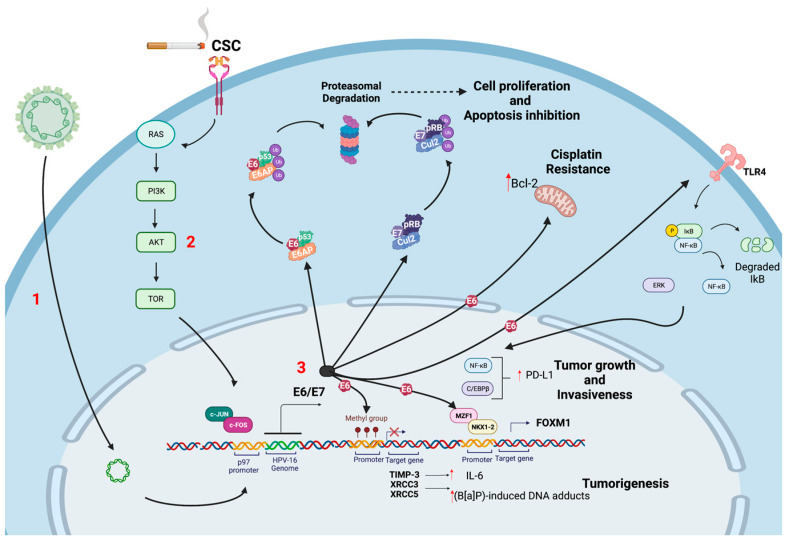
Molecular alterations in HPV-associated lung cancer. (**1**) HR-HPV infection and viral genome integration leads to E2 ORF disruption, resulting in early promoter (p97 in HPV16) activation (**2**) CSC can activate PI3K/Akt/mTOR signaling pathway, thus cooperating for increased HPV early promoter activation and E6/E7 overexpression; (**3**) HR-HPV E6/E7 oncoproteins induce p53 and pRB ubiquitination and degradation, in turn, promoting apoptosis inhibition and cell proliferation. In addition, HR-HPV E6 promote increased Bcl-2 levels and cisplatin resistance, the expression of PD-L1 through the ERK-C/EBPβ-TLR4-NF-κB signaling pathway, thus increasing tumor growth and invasiveness; induction of FOXM1 by E6 oncoprotein through the MZF1/NKX2-1 axis and the E6-induced promoter hypermethylation of the XRCC3 and XRCC5 DNA repair genes that increase tumorigenesis. Created using BioRender.

**Table 1 biology-11-01691-t001:** HPV frequency in lung cancer cases.

Tumor Types	Total	HPV (+)	(%)	HPV Genotype	HPV Gene	Methods	Country	Year	Ref
SQC	1	Condiloma	100		-	Histological changes	Finland	1972	[61]
SQC	100	-Condiloma	6		-	Histological changes	Finland	1980	[62]
		-Endophytic condyloma	4						
		-Flat type condyloma	26						
Anaplastic carcinoma in the lung	24	HPV 16 DNA	4.2		-	DNA hybridizing	Germany	1985	[63]
SQC	131	9	6.9	6, 11, 16, 18, and 30	-	ISH	Finland	1989	[64]
Squamous bronchial metaplasia, SQC	43	7	16	6, 11, 16 and 18	-	ISH	France	1990	[65]
SQC, AdC, SCLC, Large cell undifferentiated carcinoma, bronchioloalveolar carcinoma	58	7	12.1	6, 11, 16, 18, 31, 33, 35	-	ISH	USA	1992	[77]
SQC	49	7	14.3	6,11	-	PCR	Republic of China	1993	[72]
SQC, SCLC	47	0	0	-	-	PCR	Japan	1994	[73]
SQC	49	7	14.3	6, 11	-	PCR	Republic of China	1994	[75]
		5	10.2			ISH			
SQC, AdC, Neuro-endocrine cancers.	31	5	16.1	6, 11, 16	-	PCR	France	1995	[66]
SQC, AdC, SCLC, SCC, adenosquamous carcinomas	99	14	15	11, 16, 18, 33	-	PCR	Greece	1996	[67]
SQC, SCC	38	0	0	-	-	ISH	Germany	1997	[68]
SQC, AdC, SCC	50	16	32	16, 18	-	PCR, dot-blot hybridization	Republic of China	1997	[76]
SQC	34	2	6	18	-	PCR, SBH	USA	1998	[78]
SQC	52	32	69	6, 11, 16, 18	-	PCR, SBH	Greece	1998	[69]
SQC, AdC	207	18	9	6, 11, 16, 18	E6,E7	NISH	Japan	1998	[74]
SQC, AdC, oat cell carcinomas, LCC, anaplastic carcinoma	75	37	49	6, 11, 16, 18	-	PCR, ISH	Norway	1998	[71]
SQC	91	0	0	-	-	PCR, ISH	Greece	1998	[70]
SQC, AdC	185	5	2,7	16, 31, 33	-	HCA	France	2000	[79]
SQC	434	80	18	6, 11, 16, 18	E6,E7	PCR, SBH	Japan	2000	[94]
NSCLC	40	4	10	-	-	-	Poland	2001	[80]
SQC, AdC	141	77	54.6	16, 18	L1	Nested PCR, ISH	Republic of China	2001	[87]
SQC	26	3	11.5	6, 11, 16, 18	-	NISH	Turkey	2001	[81]
SQC, AdC, SCLC	40	2	5	18	L1	ISH, PCR, SBH, dot blotting	Turkey	2004	[82]
SQC	122	0	0	-	-	ISH	France	2005	[83]
SQC, AdC, Bronchioalveolar carcinomas	218	4	2	16	L1, E6	PCR	France	2005	[84]
SQC, AdC, SCC	36	10	28	16, 18, 33	L1	PCR, SBH	Japan	2006	[98]
SQC, AdC	73	23	32	16, 18	-	ISH	Republic of China	2006	[92]
NSCLC	38	8	21	16, 18, 31	E6, E7	RT-PCR	Italy	2006	[85]
NSCLC	112	58	52	16, 18, 33	E6	PCR	Korea	2007	[95]
LCLC, AdC, SQC, LCC, SCLC	141	33	26	6, 11, 16, 18, 26, 31	L1	Nested PCR	Iran	2007	[96]
AdC, SQC	69	20	29	16, 18, 31, 33, 45, 59	L1, E6	PCR, SBH, IHC	Chile	2007	[99]
AdC, SQC, LCC	78	10	13	6, 16, 31	E6, E7	PCR, RFLP	Italy	2007	[86]
AdC, SQC	313	138	39.1	16, 18	E6, E7	NISH	Republic of China	2008	[93]
AdC, SQC	217	80	37	16, 18	E6	IHC	Republic of China	2009	[88]
AdC	110	0	0	-	-	ISH	Republic of Singapore	2009	[97]
AdC, SQC	109	43	39	16, 18	L1, E6, E2	INNO-LIPA PCR, RT-PCR	Republic of China	2009	[89]
SQC	44	32	73	16, 18	-	ISH	Republic of China	2009	[90]
NSCLC	104	18	17.3	16	L1, E6, E7	RT-PCR, PCR	Republic of China	2009	[91]
AdC, SQC, LCC, SCLC	84	3	3.6	16, 18, 33	E6	PCR	Croatia	2010	[100]
AdC, SQC	57	11	19.3	6, 16, 18, 33	L1, E6, E2	RT-PCR, PCR	Japan	2010	[110]
AdC	297	0	0	16, 18, 33	L1, E1, E6, E7	Multiplex PCR, Nested PCR	Japan	2010	[111]
AdC, SQC, SCLC	59	8	13	16	L1, E6, E2	qRT-PCR, SBH	Japan	2010	[112]
AdC, SQC	30	5	16.7	16, 11	L1	PCR, DNA sequencing	USA	2010	[8]
AdC, SQC, LCC, SCLC	399	0	0	-	E6, E7	qRT-PCR	USA	2011	[128]
AdC, SQC	304	20	6.6	6, 11, 16, 18	L1	PCR, ISH	Japan	2011	[113]
AdC, SQC, SCLC	89	13	13.5	16, 30, 31, 39	E1	PCR, pyrosequencing analysis	Italy	2011	[101]
AdC, SQC	77	4	5.2	6,16	L1	Luminex-based Multimetrix	Finland	2012	[102]
AdC, SQC	100	0	0	-	L1	PCR, ISH	Italy	2012	[103]
SQC	50	9	18	6, 18	L1	Nested PCR	Iran	2013	[114]
AdC, SQC	39	16	41	16, 18	L1	PCR, ISH	Mexico	2013	[127]
AdC, SQC	170	75	44	16, 18	E6	PCR, INNO-LIPA, SBH	Republic of China	2013	[115]
AdC, SQC	336	5	1.5	16	L1	PCR, ISH	Canada	2013	[129]
AdC, SQC	200	19	9.5	16, 11	L1	qRT-PCR	Greece	2014	[104]
AdC, SQC	262	22	8.4	16,18,31	-	PCR, ISH, reverse dot blot	Republic of China	2015	[116]
AdC, SQC, SCLC	180	100	55.6	16, 18	L1	PCR and immunohistochemistry	Republic of China	2015	[117]
SQC	50	30	60	16, 18	-	ISH	Iraq	2015	[118]
AdC, SQC	196	0	0	-	E6, E7	DNA-ISH, RNA-ISH	Canada	2015	[130]
AdC, SQC, LCC, SCLC	200	5	2.5	16	L1,	Antibody assays	USA	2015	[131]
		16	8.5		E6,				
		4	2		E7				
AdC	95	27	28.4	16, 18, 33, 58	L1	PCR	Republic of China	2016	[119]
NSCLC	83	7	8.4	-	L1	Reverse hybridization	Republic of China	2016	[120]
AdC, SQC, SCLC	67	2	3	16, 53	E6, E7	NASBA	Greece	2017	[105]
AdC, SQC	132	33	25	16, 18	L1	PCR	Denmark	2017	[106]
AdC, SQC, LCC	63	33	52	16, 18	E6, E7	PCR	Brazil	2018	[132]
AdC, SQC	77	0	0	-	E7	Multiplex PCR, RT-PCR	Brazil	2019	[133]
AdC, SQC, LCC	140	13	9.3	16, 18, 35, 42, 44, 51	L1	PCR, Reverse hybridization	Republic of China	2020	[121]
AdC, SQC, LCC	80	0	0	-	E6	qPCR	Czech Republic	2020	[107]
AdC, SQC, SCLC	102	54	52.9	16	L1, E2, E6, E7	PCR, INNO-LiPA	Iran	2020	[122]
AdC, SQC, SCLC	109	56	51.4	6, 11, 16, 18, 33	L1, E,2, E6, E7	PCR, INNO-LiPA	Iran	2021	[123]
AdC, SQC	100	16	16	16	L1	PCR	Republic of China	2021	[124]
SCLC, SQC, AdC	310	183	59	-	E6, E7	qRT-PCR	Republic of China	2021	[125]
AdC, SQC, NSCLC, SCLC	41	23	56	16,18,33, 56,58	L1, E6	RT-PCR	Spain	2022	[109]

## Data Availability

Not applicable.

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
