# Peer review of "High-Risk Human Papillomavirus Infection in Lung Cancer: Mechanisms and Perspectives"

_biology, 2022, doi:10.3390/biology11121691_

Round 1
Reviewer 1 Report
The present review discusses the impact of HR-HPV genotypes on the development of Lung cancer with special focus on possible mechanisms of HPV associated lung carcinogenesis. The manuscript is well written, summarizing previous research findings concerning the association of HR-HPV infection with Lung cancer development. However, I suggest some changes.
The authors in the section “Classification, Structure, and Replication Cycle” are required to describe in more detail the function and structure of LCR (Bletsa et al., 2021, Ribeiro, et al., 2018), the function of which is essential for viral life cycle.
According to authors HPV integration seems to be a possible mechanism of lung carcinogenesis and therefore it should be highlighted in the manuscript. I suggest to describe in detail HPV DNA integration, including sites of disruption, sites of integration, molecular mechanisms of viral integration into the host genome (Tsakogiannis et al. 2017, Xu et al., 2013 Theelen et al., 2013).
Author Response
Answer: Many thanks for these observations. Additional descriptions were included in the structure and function of LCR (page 2) and regarding HPV DNA integration into the host genome (Page 8). Please note that studies addressing specific integration sites in lung cancer are lacking.
Reviewer 2 Report
The paper clearly summarizes the new evidence about correlation between lung cancers, particularly NSCLC, and HPV infection. It is divided in five parts. The second part (Human papillomavirus: Classification,Structure and Replication Cycle) is too long and detailed and should be summarized. The third part is clear and exposes the Epidemiology of HPV infection in lung cancer listing the published papers on this topic from 1996 to 2021. The final table is complete, simple and focuses on the histotypes analyzed in each paper. However, several papers published in the last year concerning this topic should be included in the review. In the fourth part, potential roles of HPV in lung carcinogenesis are exposed focusing on different carcinogenic theories highlighting the possible cooperation between HPV and tobacco in lung cancer development. In my opinion, the conclusive fifth part should be expanded including an accurate summary of the previous reported data, a comment by the authors and future prospectives.
Author Response
Answer: Many thanks for these observations and comments. It was included one paper published in 2021 and three papers published in 2022. Furthermore, a summary about the carcinogenic theories in relation a cooperation between HPV and tobacco in lung cancer was included. Finally, previous reported data and future prospective were expanded. In addition, some sections were improved with additional information and English grammar corrections.
Reviewer 3 Report
The introduction is too brief and should include rationale of the presented text.
Author Response
Many thanks for these observations. The introduction was expanded, and the rationale was added.